# Targeting undruggable carbohydrate recognition sites through focused fragment library design

Elena Shanina[1,2], Sakonwan Kuhaudomlarp[3,4,5], Eike Siebs [6,7,8], Felix F. Fuchsberger[1,2,9,10], Maxime Denis[9,10], Priscila da Silva Figueiredo Celestino Gomes[11,12], Mads H. Clausen [13], Peter H. Seeberger[1,2], Didier Rognan [11], Alexander Titz [6,7,8], Anne Imberty [3] & Christoph Rademacher [1,2,9,10 ✉]

Carbohydrate-protein interactions are key for cell-cell and host-pathogen recognition and thus, emerged as viable therapeutic targets. However, their hydrophilic nature poses major limitations to the conventional development of drug-like inhibitors. To address this short-coming, four fragment libraries were screened to identify metal-binding pharmacophores (MBPs) as novel scaffolds for inhibition of $Ca^{2+}$-dependent carbohydrate-protein interactions. Here, we show the effect of MBPs on the clinically relevant lectins DC-SIGN, Langerin, LecA and LecB. Detailed structural and biochemical investigations revealed the specificity of MBPs for different $Ca^{2+}$-dependent lectins. Exploring the structure-activity relationships of several fragments uncovered the functional groups in the MBPs suitable for modification to further improve lectin binding and selectivity. Selected inhibitors bound efficiently to DC-SIGN-expressing cells. Altogether, the discovery of MBPs as a promising class of $Ca^{2+}$-dependent lectin inhibitors creates a foundation for fragment-based ligand design for future drug discovery campaigns.

[1] Max Planck Institute of Colloids and Interfaces, Department of Biomolecular Systems, Am Mühlenberg 1, 14424 Potsdam, Germany. [2] Freie Universität Berlin, Department of Chemistry and Biochemistry, Arnimallee 22, 14195 Berlin, Germany. [3] University Grenoble Alpes, CNRS, CERMAV, Grenoble, France. [4] Department of Biochemistry, Faculty of Science, Mahidol University, 10400 Bangkok, Thailand. [5] Center for Excellence in Protein and Enzyme Technology, Faculty of Science, Mahidol University, 10400 Bangkok, Thailand. [6] Chemical Biology of Carbohydrates (CBCH), Helmholtz Institute for Pharmaceutical Research Saarland (HIPS), Helmholtz Centre for Infection Research, 66123 Saarbrücken, Germany. [7] Saarland University, Department of Chemistry, 66123 Saarbrücken, Germany. [8] German Center for Infection Research (DZIF), Hannover-Braunschweig, Germany. [9] University of Vienna, Department of Pharmaceutical Sciences, Althanstrasse 14, 1090 Vienna, Austria. [10] University of Vienna, Department of Microbiology, Immunology and Genetics, Max F. Berutz Labs, Biocenter 5, 1030 Vienna, Austria. [11] Laboratoire d'Innovation Thérapeutique, UMR 7200 CNRS-Université de Strasbourg, 67400 Illkirch, France. [12] Department of Physics, College of Sciences and Mathematics, Auburn University, 36849 Auburn, AL, USA. [13] Technical University of Denmark, Center for Nanomedicine and Theranostics, Department of Chemistry, Kemitorvet 207, 2800 Kongens Lyngby, Denmark. ✉email: christoph.rademacher@univie.ac.at

All cells are covered in a complex matrix of carbohydrates (glycans) with established roles in health and disease[1,2]. The mammalian immune system employs glycan-binding proteins (GBPs) for various processes during pathogen and tumor recognition, as well as the embryonic development[3,4]. However, many viruses (e.g. influenza[5] and SARS-CoV-2[6]), ESKAPE pathogens (e.g. *Pseudomonas aeruginosa*[7]) and tumors[8] exploit GBPs for the host invasion and to suppress the immune cell responses. Consequently, GBPs evolved as promising therapeutic targets for antimicrobials, vaccines, and as drug delivery systems to combat autoimmune diseases[9]. For example, the neuraminidase inhibitors Zanamivir (Relenza®) and Oseltamivir (Tamiflu®) are currently on the market for the treatment of influenza virus infections, which demonstrate the utility of GBP's inhibitors in anti-infectious therapy[10]. Moreover, lectins can be used as selective drug-delivery systems. This approach has been successful for Givosiran targeting hepatocytes through the asialoglycoprotein receptor (ASGPR)[11] and the development of dendritic cell-based vaccines as in the case of Dec-205 (CD205)[12]. Cumulatively, such examples increase the therapeutic interest in other clinically relevant GBPs.

In recent years, the $Ca^{2+}$-dependent C-type lectin receptors (CLRs) have emerged as therapeutic targets due to their cell-specific expression in immune cells and roles in cell signaling, self- or pathogen recognition, and antigen presentation[13]. In particular, dendritic cell-specific ICAM-3 grabbing non-integrin (DC-SIGN, CD209) and Langerin (CD207) are expressed in macrophages or dendritic cells (DCs), and a subset of skin cells called Langerhans cells, respectively[14,15]. Pathogens such as *Mycobacterium tuberculosis*[16] or viruses (e.g. HIV type-1[17], Ebola[18], SARS-CoV[19], and SARS-CoV-2[20]) use their D-mannose ($K_d$ of 3.5–6.1 mM[21]) branched carbohydrates to attach to Langerin and DC-SIGN even promoting an immune escape in some cases. Therefore, reports showing the role of DC-SIGN in promoting HIV *trans*-infection of T cells[22] or the involvement in SARS-CoV-2 uptake[23] have drawn a lot of attention to it as a potential target for antiviral therapy. Aside from pathogen recognition, Langerin is an efficient endocytic recycling receptor with established roles in immunity and tolerance[24]. Thus, both CLRs evolved as attractive targets for the design of anti-infectives, as well as for vaccine and antigen delivery to dendritic cells in immunotherapy[25].

Besides mammalian CLRs, bacterial lectins have also been proposed as viable targets for drug development, such as the opportunistic pathogens *Burkholderia ambifaria* and *P. aeruginosa* expressing BambL and LecA (PA-IL) / LecB (PA-IIL), respectively[26]. LecA and LecB attach with a remarkably high affinity to α-D-galactose ($K_d = 50 \mu M$[27]) and α-L-fucose ($K_d = 3 \mu M$[28]) containing glycans on the surface of mammalian cells causing bacterial biofilms in the lungs of immunocompromised patients. Therefore, blocking these interactions may replace or complement the antibiotic treatment of ESKAPE pathogens[29].

Developing the lectin-directed therapeutics is desirable to interfere with the bacterial[30] and viral[31] infections, as well as for cancer[32] and autoimmune diseases[33]. However, inhibitors for the orthosteric site of $Ca^{2+}$-dependent lectins have been a bottleneck for drug discovery[34]. This is not surprising given its hydrophilic nature and the high affinity for a metal cofactor such as $Ca^{2+}$ (e.g. Langerin: $K_d = 130 \mu M$[35], LecA: $K_d = 60 \mu M$[36]), which directly determines the affinity and the specificity of the carbohydrate-lectin interactions[37]. Therefore, the flexibility and electronic structure of the metal-centered complexes further complicate shaping metal-fragment interactions in 'classical' drug design[38]. This overall rendered this target class challenging and contributed to the vast underrepresentation of lectin inhibitors in the drug space[39].

Recent advances towards the development of lectin inhibitors focus on mimicking the natural ligands of lectins (glycomimetics) through the design of mono- and multivalent oligosaccharides with various conjugates (glycopeptides, glycoclusters, and glycopolymers)[40]. However, they pose a pharmaceutical challenge such as routes of administration and possible side effects[39]. Several carbohydrate-based glycomimetics are available for CLRs[41,42] and bacterial lectins[29,43,44], but only a few non-carbohydrate drug-like lectin inhibitors are reported[45,46]. Previously, fragment-based drug design (FBDD) was successfully applied for assessing the druggability of CLRs[47]. However, fragments likely to interfere with primary in vitro screening assays (PAINS) were excluded from the general fragment libraries as their presence can partially or fully contradict the screen[48]. Therefore, a target-directed FBDD campaign using metal-binding pharmacophores (MBPs) has been proposed[49]. Certainly, drugs on the market as captopril[50] and suberanilohydroxamic acid (Zolinza)[51] demonstrate the success of this strategy. Interestingly, many clinically relevant proteins in the genome as lectins require a metal ion to maintain their stability and activity, but MBPs have not been considered for non-metalloenzymes[49].

Here, we show that metal-dependent lectins are more druggable than previously anticipated. To address this, we screened four fragment libraries identifying MBPs as promising scaffolds. To demonstrate the potential of MBPs in inhibition of carbohydrate-protein interactions, we studied their effect on clinically relevant $Ca^{2+}$-binding lectins (DC-SIGN, Langerin, LecA, and LecB), whereas a metal-independent lectin BambL served as a control (Fig. 1a). Employing NMR[52], surface plasmon resonance (SPR), X-ray crystallography, as well as fluorescence polarization (FP[53]) assays, we explored the structure-activity relationships of several MBPs to further improve lectin binding and selectivity. Finally, the activity of MBPs in a physiologically relevant cellular environment was assessed in a cell-based fragment screen assay (CellFy)[54].

## Results

**Metal-binding pharmacophores target $Ca^{2+}$-dependent lectins.** The druggability of four $Ca^{2+}$-dependent lectins was assessed and compared to a metal-independent lectin BambL using a virtual and two experimental: 3F Fsp³-rich and diversity-oriented (general), libraries (Fig. 1b), as described in the Supplementary Results and Discussion (Figure S1). Despite the limitations of in silico approaches, $^{19}F$ NMR screening of the general library revealed the druggability of $Ca^{2+}$-dependent lectins LecA and LecB, where MBPs were identified as the most potent drug-like molecules (Figures S2 and S3).

To demonstrate the potential of MBPs in targeting the carbohydrate-binding site of $Ca^{2+}$-dependent lectins, we subjected 142 commercial MBP fragments for the binding studies with one non-$Ca^{2+}$-(BambL) and four $Ca^{2+}$-dependent lectins (LecA, LecB, Langerin and DC-SIGN). In the $^{19}F$ NMR screening of the MBP library, the chemical shifts of many $^{19}F$-labeled fragments were perturbed in the presence of $CaCl_2$ alone, whereas even larger CSPs were observed in the presence of both $CaCl_2$ and the $Ca^{2+}$-dependent lectins as shown on the example of LecA binding to **1s-1v** (Fig. 1c), which allowed us to exclude direct fragment-metal interactions as a major contribution. Notably, the MBP library improved the hit rates for lectins with one (LecA: 54.2%, Langerin: 62.8%), two (LecB: 42.8%) and three $Ca^{2+}$ ions (DC-SIGN: 42.5%), but not for the metal-independent lectin BambL (7.7%) compared to 3F Fsp³-rich and general libraries (Fig. 1d).

Next, we validated $^{19}F$ NMR hits from the general and MBP libraries for binding to $^{15}N$-labeled LecA, LecB and the carbohydrate recognition domain (CRD) of DC-SIGN in $^{1}H$-

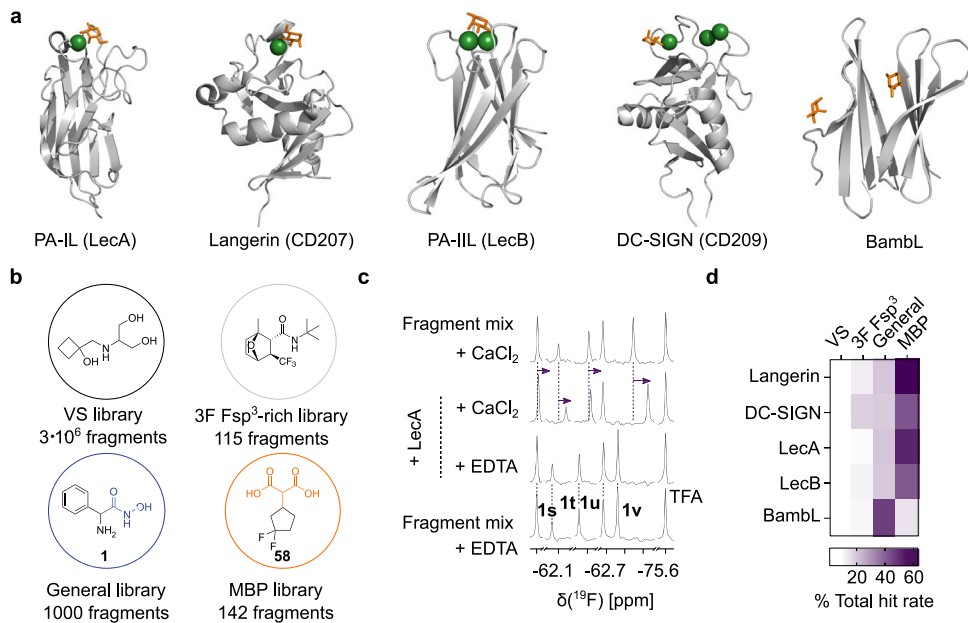

**Fig. 1 Druggability assessment of metal-dependent lectins. a** Cartoon representation of monomer crystal structures showing the carbohydrate-recognition domain (CRD) bound to the monosaccharides (*orange*) in a $Ca^{2+}$-dependent manner (*green spheres*): LecA (D-galactose, PDB: 4CP9), Langerin (D-mannose, PDB: 3P5D), DC-SIGN (D-mannose, PDB: 1SL4), LecB (L-fucose, PDB: 5A70). BambL does not require $Ca^{2+}$ for carbohydrate binding (L-fucose, PDB: 3ZZV). **b** All targets were screened computationally and experimentally using 3F $Fsp^3$, general and MBP libraries. **c** Shown are $^{19}F$ $T_2$-filtered NMR spectra of a fragment mixture of the MBP library in the presence of 5 mM EDTA alone or with 20 μM LecA followed by competition with 30 mM $CaCl_2$. The fragments undergoing a chemical shift perturbation (CSP) above 0.01 ppm (*violet arrow*) in the presence of LecA and $CaCl_2$ indicated a $Ca^{2+}$-dependent binding. **d** Heat map shows the percentage values (%) of total hit rates of four libraries screened against five lectins revealing that MBP library improved the hit rates for $Ca^{2+}$-binding lectins (37–50%).

$^{15}N$ HSQC/TROSY NMR experiments, identifying hydroxamic (**1**) and malonic (**58**) acids as potent scaffolds for targeting $Ca^{2+}$-dependent lectins (LecA LecB and DC-SIGN). Further, both scaffolds were subjected to explorative structure-activity relationship (SAR) studies aiming to demonstrate the $Ca^{2+}$-dependency and selectivity of the scaffolds for the $Ca^{2+}$-dependent lectins.

**Hydroxamates as $Ca^{2+}$-dependent inhibitors of LecA.** To characterize the interaction of LecA with the hydroxamate **1** and to enhance its potency, we initiated a SAR study employing five biophysical assays. A full discussion can be found in the Supplementary Results and Discussion. Briefly, we ranked 49 analogs of **1** using TROSY NMR, where we quantified the changes in the chemical shift perturbations (CSPs) of $^{15}N$ LecA in the presence of the **1** analogs and MeGal as a positive control (Figure S4a). The derivatives promoted CSPs in $^{15}N$ LecA similar to MeGal and **1** as shown in the case of **35** (Fig. 2a, b), suggesting that hydroxamates targeted the orthosteric site of LecA. Moreover, none of the marketed metalloproteinase inhibitors (**47**, **49**, **50**) bound to $^{15}N$ LecA (Figures S4b-c), which indicated the presence of functional groups in the marketed drugs that sterically prevented a beneficial coordination of $Ca^{2+}$ in LecA. Cumulatively, we identified 18 analogs of hydroxamate **1** targeting LecA with the potentially higher affinities.

In parallel, we validated TROSY NMR results using SPR, competitive $^{19}F$ NMR, FP assay and protein-observed $^{19}F$ (PrOF) NMR. Given a weak affinity of hydroxamates towards LecA, SPR did not allow ranking of the derivatives (Figure S5). Therefore, we designed a competitive $T_2$-filtered $^{19}F$ NMR experiment using the hydroxamate derivative (**5**) as a $^{19}F$ spy molecule. To prove the utility of this assay, we evaluated the $Ca^{2+}$-dependency and selectivity of the hydroxamate-LecA interaction and compared it to other metal-dependent lectins, i.e. LecB, DC-SIGN and

Langerin (Fig. 2c). To test the $Ca^{2+}$-dependency of the interactions of **5** with lectins, a $T_2$-filtered $^{19}F$ NMR spectrum of **5** was recorded with 2 mM EDTA, 10 mM $CaCl_2$ alone and with 10 μM lectins added. Both Langerin and DC-SIGN showed a $Ca^{2+}$-independent binding to hydroxamates (Figure S6), whereas only **5**-LecA interactions required $Ca^{2+}$, demonstrating the $Ca^{2+}$-dependency and early selectivity of the hydroxamate-LecA interaction. Consequently, we used this assay to rank the derivatives as discussed in the Supporting Information. Briefly, we observed a higher competition of **35** over the reporter **5** and other analogs suggesting **35** as the most potent hydroxamate derivative in this assay (Figure S7).

Next, we investigated the inhibitory properties and derived the affinities ($K_d$) of hydroxamates, which were active in NMR. In the FP assay the strongest inhibition compared to natural ligands of LecA, MeGal and GalNAc, was observed with linear **35** and cyclic **20** (Fig. 2d, Table 1)[53]. Following, PrOF NMR using LecA labeled with 5-fluorotryptophans (5FW) was employed. Since W42 is located in the orthosteric site of LecA, the binding and affinities of weak ligands could be determined[36]. As result, the perturbation of W42 in the presence of hydroxamates confirmed its binding to the orthosteric site of LecA (Fig. 2e and S8a, Table S4). Moreover, changes in W42 were used to estimate the affinities ($K_d$) of hydroxamates for LecA, as shown on example of **35** ($K_d$ (**35**) = 4.6 ± 0.9 mM, Fig. 2e and S8b-c, Table S3). The analogs showed a similarly poor affinity compared to **35**. Due to the small size of the fragments, there are only a limited number of the interactions they can engage in. However, they may bind tightly enough relative to their size and number of heavy atoms (HAs)[55]. Ligand efficiency (LE) is the key measure relating the potency of a fragment to the number of HAs and thus, would allow the development of drug-like molecules with enhanced binding modes for LecA[56]. Therefore, we selected only fragments with the highest affinity, LE and the % of inhibition values (Table 1).

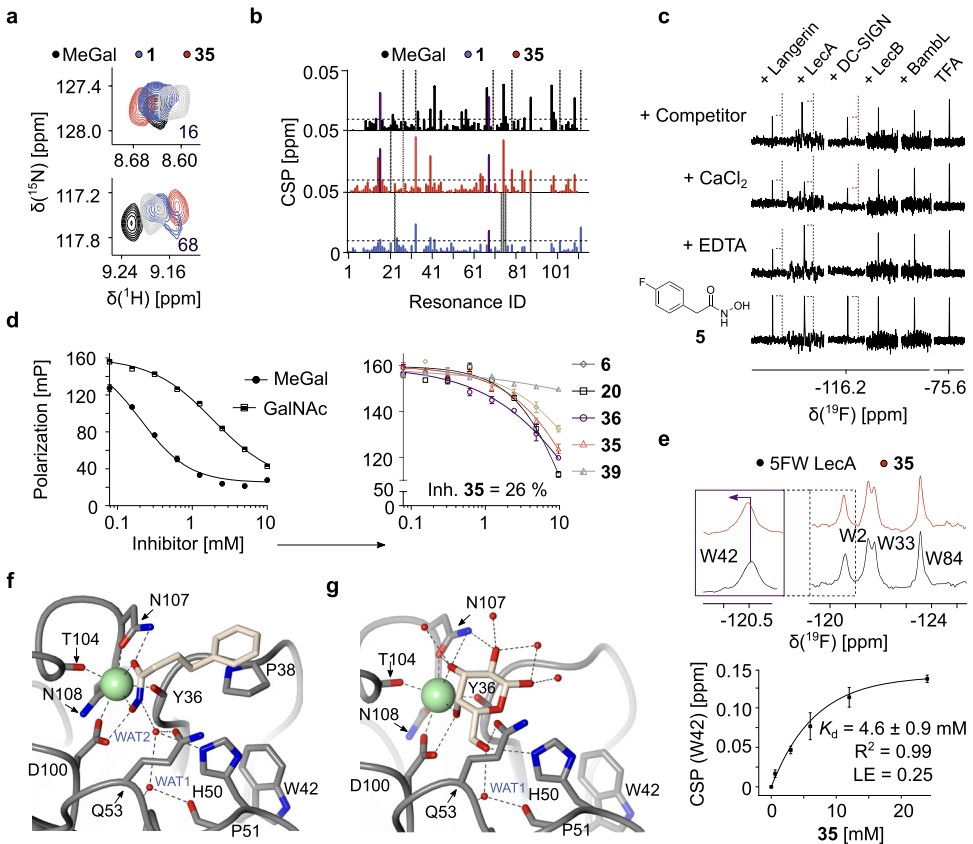

**Fig. 2 Hydroxamates as drug-like inhibitors of PA-IL (LecA). a** Examples of the residues in the carbohydrate-binding site of $^{15}$N LecA in TROSY NMR in the presence of DMSO (*gray*), 3 mM **1** (*blue*), **35** (*red*) or 1 mM methyl α-D-galactoside (MeGal, *black*). **b** Quantitative analysis of the chemical shift perturbations (CSPs > 0.01 ppm, *dashed line*) and changes in the peak intensity (*dotted line*) promoted by **1**, **35** and MeGal. The resonances 16 and 68 were highlighted (*violet*) showing hydroxamates perturbed the residues similarly to MeGal. **c** Shown is a Ca$^{2+}$-dependent binding of 0.01 mM LecA to **5** (*dashed line*) in the competitive $^{19}$F T$_2$-filtered NMR, as evident by a decrease and recovery of the spy **5** with Ca$^{2+}$ (10 mM CaCl$_2$) and without (2 mM EDTA), respectively. Moreover, **5**-LecA interaction showed a higher selectivity over other non- and Ca$^{2+}$-dependent lectins. **d** Competitive binding assay based on fluorescence polarization with MeGal and GalNAc (*left*) as positive controls for strong and weak ligands, respectively. The hydroxamates **6**, **20**, **35**, **36** and **39** (right) demonstrated LecA-dependent inhibitory properties. Inhibition values of compounds were derived at 10 mM and referenced to MeGal (16 h). **e** Shown is PrOF NMR of 0.15 mM 5FW LecA with 3 mM **35** (*upper panel*) binding to the carbohydrate pocket as evident by a CSP of W42. One-site binding model of **35** (*bottom panel*) used to derive the affinity (K$_d$) by following CSPs of W42. The error bars are showing the mean (n = 3). The K$_d$ values were used to calculate LE. **f** Crystal structures show the interactions between LecA and **35** (PDB: 7FJH) and (**g**) galactose (PDB: 1OKO).

Crystal structure of LecA in complex with **35** at 1.8 Å resolution in space group P2$_1$2$_1$2$_1$ provided the first evidence for the interaction between a hydroxamate and a nonenzyme metal-binding protein. The electron density corresponding to the full structural model for **35** was detected at the carbohydrate-binding sites in subunits A and D of LecA (Fig. 2f and S9a-b). Comparing the structures of LecA-**35** and LecA-galactose complexes illustrated the galactose-mimicking properties of **35** (Fig. 2g). In the LecA-**35** complex, two oxygen atoms in the hydroxamic acid functional group coordinate the Ca$^{2+}$ ion *via* bidentate chelation, which is also typical for interactions between hydroxamate-based inhibitors and metal ions in metalloenzymes[49]. The two oxygen atoms form hydrogen bonds with N107 and D100, mimicking OH3 and OH4 of galactose. A water molecule (WAT2) is in contact with the nitrogen atom in the hydroxamic acid *via* a hydrogen bond. WAT2 mimics the role of galactose OH6 by forming hydrogen bonds with the side chains of H50, Q53, the main chain oxygen atom of P51 and water WAT1. Notably, the terminal benzyl ring forms CH-π interactions with P38, an interaction not observed for other LecA-targeting glycomimetics (Figure S9c)[57].

Taken together, we report the first evidence of hydroxamates targeting carbohydrate-protein interactions. Moreover, we

conducted several experiments highlighting the selectivity of hydroxamates towards LecA over LecB and mammalian Ca$^{2+}$-dependent lectins (Langerin/DC-SIGN). Therefore, we believe hydroxamates are promising molecules for the design of drug-like inhibitors against bacterial infections.

**Malonates target lectins with multiple Ca$^{2+}$ ions.** In the $^{19}$F and $^1$H-$^{15}$N HSQC/TROSY NMR screenings of the MBP library, we discovered malonate **58** as a potent scaffold for targeting the Ca$^{2+}$-dependent lectins LecA, LecB and DC-SIGN. To prove the utility of a target-oriented MBP-FBDD approach, we aimed to unravel its interactions with lectins having one, two, and three Ca$^{2+}$ ions in or close to the orthosteric site. Here, BambL served as control for a metal-independent lectin expected not to interact with MBPs such as malonates. For this, the malonate **58** was subjected to docking and an initial SAR study using commercial analogs (Fig. 3a, Table S6). The results of both studies are discussed in detail in the Supplementary Results and Discussion. Briefly, docking simulation proposed that the malonate **58** could coordinate one or multiple Ca$^{2+}$ ions in LecA, DC-SIGN and LecB, respectively (Fig. 3a, S10–S12). Therefore, we investigated the Ca$^{2+}$-dependency and selectivity of malonate-lectin

interactions. Taking advantage of a C-F bond in α-position of the malonic acid group of **61**, we used it similar to **5** as a $^{19}F$ spy in a competitive $^{19}F$ $T_2$-filtered NMR experiment (Fig. 3b). We observed **61** binding to all lectins except for Langerin in the presence of 10 mM $CaCl_2$ suggesting that malonates target the orthosteric sites of lectins in a $Ca^{2+}$-dependent manner. To rule

out an off-target effect, we added a competitor (LecA: 10 mM MeGal, LecB/BambL: 10 mM MeFuc, and DC-SIGN/Langerin: 30 mM D-mannose) expecting the $^{19}F$ peaks to recover if malonates target the orthosteric site of $Ca^{2+}$-dependent lectins. Indeed, the competitors displaced **61** in LecA, LecB and DC-SIGN. On the other hand, the $^{19}F$ peak intensities remained unchanged in Langerin and BambL, which was probably due to **61** binding to the secondary druggable sites[58–60]. Cumulatively, our results proposed that malonates interact with the orthosteric sites of metal-dependent lectins LecA, LecB and DC-SIGN in a $Ca^{2+}$-dependent manner.

Next, we aimed to confirm our observations for LecA and LecB and investigated whether the number of $Ca^{2+}$ ions in the orthosteric site plays a role for the interaction with malonates. For LecA, PrOF NMR with 5FW-labeled lectin was used to gain information on the binding site of **58** analogs. Similar to hydroxamates, **58** perturbed W42 in PrOF NMR demonstrating its binding to the orthosteric site of 5FW LecA (Fig. 3c). Subsequently, we aimed to rank the impact of 13 analogs of **58** on W42 to prioritize scaffolds for future fragment evolution. The compounds with an acetic acid group (**63** and **70**, Table S6) did not interact with 5FW LecA, which supported our docking result (Figures S10a-b), whereas the scaffold **61** promoted the strongest perturbation of W42 (Figures S10c-d). Altogether, malonates (**58** and **61**) interact with the orthosteric site of LecA bearing one $Ca^{2+}$ ion, similar to hydroxamates.

For LecB as a lectin with two $Ca^{2+}$ ions, a similar tendency in malonate binding to the orthosteric site was observed in TROSY NMR. Here, we observed similar changes in residues of $^{15}N$ LecB resonances with **58** compared to the positive control MeFuc, whereas the binding was in a $Ca^{2+}$-dependent manner (Fig. 3d, S13–S14). Therefore, we determined the affinities ($K_d$) and LE of

**Table 1 Overview of hydroxamate 1 analogs for targeting LecA.**

| Compound | | $K_d$ [mM] | LE [kcal mol$^{-1}$ HA$^{-1}$] | Inhibition [%] |
|---|---|---|---|---|
| **1** | (phenyl with C(=O)N(H)-OH, NH₂) | 7.2 ± 1.4 | 0.25 | 6* |
| **6** | (cyclohexyl with CH₂C(=O)N(H)-OH) | 4.4 ± 0.6 | 0.30 | 21 ± 1 |
| **20** | (quinazolinone with N-OH, CH₃) | 4.5 ± 0.2 | 0.26 | 33 |
| **35** | (phenyl with chain C(=O)N(H)-OH) | 4.6 ± 0.9 | 0.26 | 26 ± 1 |
| **36** | (methylphenyl with chain C(=O)N(H)-OH) | 3.6 ± 2.2 | 0.25 | 35 ± 3 |

Inhibition [%] compared to MeGal at 10 mM (16 h).
*Measured at 4 mM

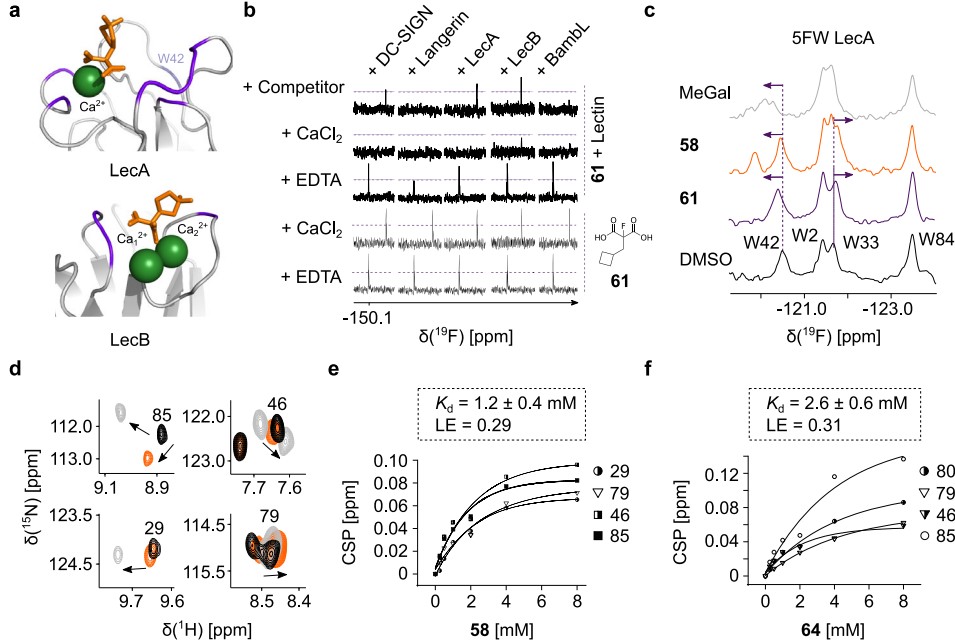

**Fig. 3 Malonates target lectins with one or multiple calcium ions. a** Docking poses of **58** with one and two $Ca^{2+}$ in or near the carbohydrate-binding site of LecA and LecB (PDB: 4CP9, 1OXC). The residues proposed to interact with lectins were highlighted (*violet*). **b** $^{19}F$ $T_2$-filtered NMR using the analog of **58** (**61**) was used to assess the $Ca^{2+}$-dependency and selectivity of the malonate-lectin interaction, where **61** bound to 10 mM $CaCl_2$ given a perturbation of fluorine resonance. All lectins interacted with **61** in a $Ca^{2+}$-dependent manner. The presence of competitors: 10 mM MeGal (LecA), 10 mM MeFuc (LecB/ BambL) and 30 mM D-mannose (DC-SIGN/Langerin) displaced **61** in LecA, LecB and DC-SIGN ECD verifying that malonates target the carbohydrate-binding site. **c** Shown is PrOF NMR of 0.15 mM 5FW LecA bound to 2 mM **58** or **61** as evident by CSPs of W42 and W33 (*arrow*). DMSO and MeGal served as negative and positive controls, respectively. (**d**) Examples of $^{15}N$ LecB residues in TROSY NMR alone (*black*) and with 2 mM **58** (*orange*) or MeFuc (*gray*). **58** perturbed the resonances 85, 46, 29 and 79 similarly to MeFuc. **e** and **f** One-site binding model was applied to derive the $K_d$ values of **58** and **64** by following CSPs of the resonances perturbed similar to MeFuc.

**Table 2 Overview of malonate 58 analogs to target LecB and DC-SIGN.**

| Compound | | DC-SIGN CRD | | LecB | |
|---|---|---|---|---|---|
| | | $K_d$ [mM] | LE [kcal mol$^{-1}$ HA$^{-1}$] | $K_d$ [mM] | LE [kcal mol$^{-1}$ HA$^{-1}$] |
| 58 | | 1.2 ± 0.5 | 0.28 | 1.2 ± 0.4 | 0.29 |
| 59 | | 1.2 ± 0.4 | 0.31 | 2.7 ± 0.6 | 0.28 |
| 62 | | 1.5 ± 0.7 | 0.26 | 1.5 ± 0.2 | 0.27 |
| 64 | | 2.6 ± 0.9 | 0.29 | 2.6 ± 0.6 | 0.31 |
| 66 | | 1.3 ± 0.4 | 0.33 | 1.6 ± 0.4 | 0.33 |
| 67 | | 1.2 ± 0.5 | 0.33 | 1.6 ± 0.3 | 0.31 |
| 69 | | 1.0 ± 0.6 | 0.37 | 1.7 ± 0.6 | 0.36 |

**58** analogs using TROSY NMR, which were in a low mM range (**58**: $K_d$ = 1.2 ± 0.4 mM, Fig. 3e). Interestingly, all structural derivatives of **58** gave a rather flat SAR and showed comparable affinities for LecB (Table 2, Figure S15), which is discussed in detail in the Supporting Information. Notably, only **64** had a decrease in affinity (**64**: $K_d$ = 2.6 ± 0.6 mM, Fig. 3f) indicating the role of an electronegative group in binding to LecB. This observation supported our docking study, where CF$_2$ group of **58** was predicted to interact with the protein surface through T98 using a hydrogen bond (Figures S11a-b). However, due to the lack of $^{15}$N LecB protein backbone assignment, co-crystallization studies are ongoing to support it.

Taken together, malonates interacted with lectins with one (LecA) and two (LecB) Ca$^{2+}$ ions in the orthosteric site unlike hydroxamates coordinating only one Ca$^{2+}$ ion in the orthosteric site of LecA. Moreover, our SAR study with LecA and LecB suggested that the selectivity of malonates is tunable as **61** had a stronger tendency to bind LecA, whereas **58** showed the preference for LecB. Finally, malonates have a higher tendency to bind to secondary sites in allosteric lectins, such as Langerin and BambL. Cumulatively, malonates offer a potential scaffold for design of orthosteric and allosteric inhibitors.

**Malonates to design inhibitors of DC-SIGN (CD209).** To investigate the ability of malonates to bind lectins with three Ca$^{2+}$ ions, we probed malonate—DC-SIGN interaction using protein-observed $^{15}$N HSQC NMR and a cell-based assay (cellFy)[54]. A full discussion on the SAR study of **58** and DC-SIGN can be found in the Supplementary Results and Discussion (Figure S16). Briefly, malonate **58** perturbed the resonances in the EPN motif coordinating Ca$^{2+}$ (site 1) and D367, whereas L321 and E324 near Ca$^{2+}$ (site 2) and Ca$^{2+}$ (site 3) showed weaker effects (Fig. 4a, b).

Quantitative analyses of the chemical shift perturbations (CSPs) caused by **58** revealed similar CSPs compared to the positive control D-mannose, suggesting that **58** mimics the carbohydrate binding to DC-SIGN (Fig. 4c, d). In particular, the interaction of **58** with the Ca$^{2+}$(site 1)-binding site of DC-SIGN occurred through N365, N344 and F313 being in agreement with our docking predictions. We then derived the affinities ($K_d$) and LE values of **58** analogs for DC-SIGN CRD using $^{15}$N HSQC NMR. The analogs of **58** showed a similar affinity (**58**: $K_d$ = 1.2 ± 0.5 mM, Table 2, Fig. 4e and S17) and thus, three scaffold groups were defined as interchangeable (**58**, **62** and **69**). Similar to LecB, the compounds with an electronegative group on the ring (**58**, **62**, **63** and **67**) were predominant and thus, in agreement with the predicted F313 interaction of DC-SIGN with CF$_2$ group in **58**. Compared to LecA and LecB, a methyl group in **59** was well tolerated in DC-SIGN CRD (Fig. 4f). Therefore, this position is potentially suitable for future fragment growing to gain malonate specificity towards DC-SIGN. Together, both computational and experimental data demonstrated that malonates could target the Ca$^{2+}$ (site 1) binding site of DC-SIGN similarly to D-mannose.

To assess the activity of **58** in a physiologically more relevant cellular environment, we explored the ability of malonates to inhibit the carbohydrate binding in a cell-based system (cellFy)[54]. For this, we assessed the binding of **58** to DC-SIGN$^+$ and Langerin$^+$ Raji cells by flow cytometry (Fig. 4g). Indeed, we observed dose- and DC-SIGN-dependent binding of **58** to DC-SIGN$^+$ cells as well as negligible cytotoxicity at low concentrations. Most importantly, malonate **58** specifically bound to DC-SIGN$^+$ cells and not to Langerin-expressing cells, although both lectins share a common recognition motif called EPN. Together, our cellFy data demonstrated that malonates specifically inhibit DC-SIGN-carbohydrate interactions in the cellular context similar to D-mannose.

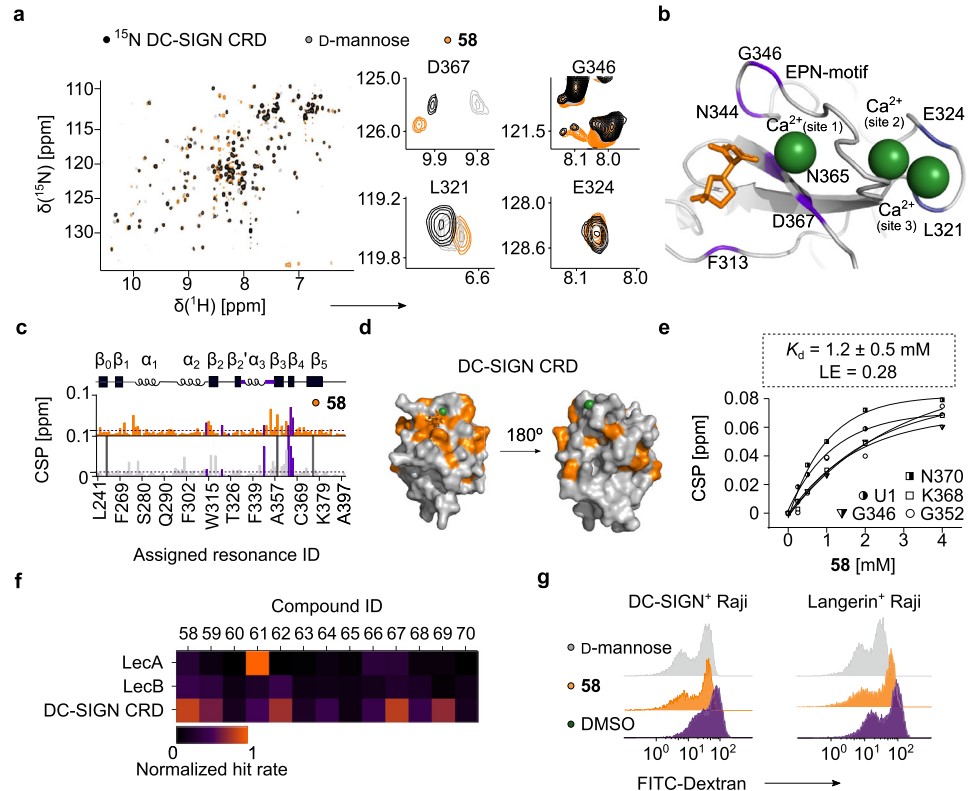

**Fig. 4 Malonates as inhibitors of DC-SIGN (CD209). a** Protein fingerprint of $^{15}$N DC-SIGN CRD in $^{15}$N HSQC NMR alone (*black*) and with 2 mM **58** (*orange*) or 30 mM D-mannose (*gray*). Similar to D-mannose, **58** perturbed the resonances D367, G346 and L321, but not E324 being in the Ca$^{2+}$ (site 1) and Ca$^{2+}$(site 1)/Ca$^{2+}$(site 3)-binding sites, respectively. These residues are highlighted in (**b**), which shows a docked **58** in the Ca$^{2+}$ site 1 of DC-SIGN CRD (PDB: 2XR5). **c** Quantitative analysis of chemical shift perturbation (CSP) of $^{15}$N DC-SIGN CRD in the presence of **58** and D-mannose. Resonances D367, G346, L321 and E324 are highlighted (*violet*). A schematic representation of DC-SIGN CRD is shown on top. Malonates perturbed resonances similarly to D-mannose, i.e. in the long loop locating the EPN motif (*violet*). **d** The CSPs promoted in $^{15}$N DC-SIGN CRD by **58** were transferred to DC-SIGN CRD (PDB: 2XR5). **e** One-site binding model of **58** to DC-SIGN CRD derived by following CSPs of the resonances perturbed in the long loop and β$_4$ sheet. **f** Heat map shows the 1:0 hit rate of **58** analogs for LecA, LecB and DC-SIGN normalized to the effect of MeGal, MeFuc and D-mannose, respectively. **g** Shown are flow cytometry histograms of DC-SIGN$^+$ and Langerin$^+$ Raji cells bound to FITC-dextran in the presence of DMSO or D-mannose used as a negative and positive controls in a cell-based (cellFy) assay, respectively. **58** inhibited FITC-dextran binding to DC-SIGN similarly to D-mannose, but not to Langerin.

Together, our SAR study demonstrated that the selectivity of malonates is tunable. Moreover, the malonate scaffold can inspire the design of lectin inhibitors with one, two and three Ca$^{2+}$ ions, as shown exemplarily for LecA, LecB and DC-SIGN, respectively. Thereby, **61** had a stronger tendency to interact with LecA, whereas LecB and DC-SIGN preferred **58** and **62** scaffolds, and the observation with **59** is valuable to gain malonate specificity towards DC-SIGN.

## Discussion

Interfering with GBP interactions has a high therapeutic potential. However, identifying a suitable starting point for the development of lectin inhibitors is challenging. Despite that several mono- and multivalent carbohydrate-based glycomimetics are available for lectins[40], poor pharmacokinetic properties of the most potent multivalent inhibitors are a large drawback for the clinical approval. This further underscores the attractiveness of drug-like noncarbohydrate lectin inhibitors. Applying fragment screening, we approached this challenge focusing on metal-dependent lectins. As result, we identified metal-binding pharmacophores (MBPs) as novel scaffolds for targeting Ca$^{2+}$-dependent carbohydrate-protein interactions.

Recently, we reported the molecular basis of a catechol binding to LecA, where two hydroxyls coordinated a Ca$^{2+}$ ion in the orthosteric site of LecA and Langerin[45]. Here, we identified hydroxamic acids as additional drug-like molecules to target the Ca$^{2+}$-bearing orthosteric site of LecA. As hydroxamates are the most widely used inhibitors of various metalloenzymes, to the best of our knowledge this is the first report demonstrating their interaction with non-metalloenzymes. The SAR study revealed a sterically optimal presentation of the hydroxamic acid group. However, the potencies of hydroxamates are certainly lower than those observed for metalloenzymes, and still have to be improved. Nevertheless, hydroxamates offer a promising selectivity towards LecA over other Ca$^{2+}$-dependent lectins. This knowledge will contribute to the design of specific hydroxamate-based inhibitors for LecA, which have the potential to interfere with *P. aeruginosa* biofilm integrity and increase susceptibility to antibiotic treatment of immunocompromised patients.

The improved druggability rates for the orthosteric site of the Ca$^{2+}$-dependent lectins with catechols and hydroxamates encouraged us to initiate a target-oriented MBP-FBDD campaign. Employing MBP library, we have shown a fourfold improved hit rates for lectins with one (LecA and Langerin) and multiple Ca$^{2+}$ ions (DC-SIGN and LecB) in the orthosteric site. Hereby, a malonate scaffold **58** was identified as a potent drug-like molecule for targeting lectins with one or multiple Ca$^{2+}$-ions. Notably, this malonate showed low cytotoxicity (up to 10 mM) and selectivity for DC-SIGN$^+$, but not Langerin$^+$ Raji cells in vitro using a cell-based assay (cellFy). Moreover, using commercially available analogs of malonate **58**, we demonstrated that tuning the

selectivity of malonates towards lectins with one to multiple $Ca^{2+}$ ions was possible. Aside from the $Ca^{2+}$-dependent binding of a common malonic acid group, we discovered an additional interaction of **58**'s electronegative $CF_2$ group with the protein surfaces of DC-SIGN and LecB in HSQC/TROSY NMR. This secondary interaction was not observed for LecA, which demonstrated a preference for a malonate with a different scaffold (**61**). We believe our discovery will assist the design of malonate-based lectin inhibitors.

Taken together, metal-coordinating fragments are the most prominent drug-like molecules for targeting the metal-dependent lectins. Since the most potent hydroxamate (**35**) and malonate (**58**) scaffolds are still small with 179 and 208 Da, respectively, both serve as suitable starting points for the fragment evolution, which could lead to the first drug-like inhibitors for the orthosteric sites of mammalian and bacterial $Ca^{2+}$-dependent lectins.

## Methods

**Materials**. The carbohydrates D-mannose (CAS: 3458-28-4) and methyl α-D-galactoside (MeGal, CAS: 3396-99-4) were purchased from Sigma-Aldrich Chemie GmbH, whereas methyl α-L-fucoside (MeFuc, CAS: 14687-15-1) was from Biosynth-Carbosynth (UK). Commercial analogs (Table S3, compounds **3-5**, **9-35**, **46-50**) of hydroxamic and malonic acids (Table S6) were purchased from Life Chemicals Europe GmbH (Germany), Otava Chemicals (Lithuania) or Key-Organics (UK). The non-labeled or $^{15}$N-labeled LecA[61], LecB[62] and BambL[63] were purified in soluble form as reported previously. Human $^{15}$N-labeled Langerin CRD and non-labeled ECD or DC-SIGN CRD and non-labeled ECD constructs were expressed in inclusion bodies and prepared as described before[35,58].

**Virtual screening**. The detailed protocol including the list of analyzed LecA (Table S1) and LecB (Table S2) crystal structures used for virtual screening was described in Supplementary Materials and Methods.

**Fragment libraries**. The 3F and general $^{19}$F fragment libraries were prepared as reported previously[47,64]. MBP library of 142 commercially available fragments purchased from Otava Chemicals (Lithuania) was prepared. MBP fragments were chosen from the 'Chelator Fragment Library' based on the presence of a fluorine atom and various chelating groups represented by picolinic acids, pyrimidines, hydroxypyrones, hydroxypyridinones, sulfonamides, β-diketones, salicylic and hydroxamic acids. Fragments were subjected to a quality control for solubility and purity in $^{1}$H and $^{19}$F NMR yielding 98 fluorinated and 9 nonfluorinated fragments. Following, we combined fluorinated MBP fragments in mixtures of 30 to 32 fragments at 100 μM in a two-fold concentrated $^{19}$F screening buffer (25 mM in Tris-HCl pH 7.8, 150 mM NaCl, 100 μM TFA and 20% $D_2O$) and stored at −20 °C upon use.

**Chemical synthesis**. Hydroxamate derivatives were prepared as well as analyzed for the identity and purity by NMR as described in Supplementary Materials and Methods and Supplementary Note 1, respectively.

**NMR studies**. All NMR measurements were performed on a Bruker Ascend™700 (AvanceIII HD) spectrometer equipped with a 5 mm TCI700 CryoProbe™ in 3 mm tubes (Norell S-3-800-7).

**Fragment screening**. NMR screening of fluorinated and non-fluorinated fragments from the 3F, general and MBP libraries was performed in $^{19}$F NMR and $^{1}$H-$^{15}$N HSQC/TROSY NMR, respectively. For $^{19}$F NMR, two samples containing 1:1 fragment mixture and 10 μM protein in TBS/EDTA buffer (25 mM in Tris-HCl pH 7.8, 150 mM NaCl, 5 mM EDTA) or TBS/EDTA buffer alone were prepared and followed by addition of 10 mM $CaCl_2$ and 1 mM MeGal (LecA), 1 mM MeFuc (LecB/BambL) and 30 mM D-mannose (DC-SIGN/Langerin). Two separate $^{19}$F NMR spectra were recorded at 298 K for $CF_3$ and CF groups with 32 and 64 scans, a spectral width of 100 ppm, a transmitter offset at −50 and −150 ppm, acquisition time of 2 s and 1 s relaxation time, respectively. $^{19}$F $T_2$-filtered (CPMG) spectra were recorded using a CPMG pulse sequence with a 180° pulse repetition rate of 384 ms using the same acquisition and relaxation times with 64 and 256 scans for $CF_3$ and CF compounds, respectively. Data was acquired without proton decoupling. All spectra were referenced to the internal standard trifluoroacetic acid (TFA) at −75.6 ppm and analyzed in MestReNova 11.0.0 (Mestrelab Research SL) for the changes in peak intensity and chemical shift perturbations in the presence of protein. For $^{19}$F NMR spectra, chemical shift changes of 0.01 ppm or intensity changes between 25 and 50 % were defined as 'high' and 'low' confidence hits, respectively. For $^{19}$F $T_2$-filtered NMR spectra, intensity changes of 20 and 50% or

more than 50% were defined as 'low' and 'high' confidence hits, respectively. Fragments that bound proteins in the presence of 10 mM $CaCl_2$ were used to derive total hit rates.

**$^{1}$H-$^{15}$N HSQC and TROSY**. The $^{1}$H-$^{15}$N TROSY pulse sequence *trosyf3gpphsi19* with 128 increments and 32 scans per increment was applied for 0.15 mM $^{15}$N LecA and 0.07 mM LecB in HBS/$Ca^{2+}$ buffer (20 mM HEPES pH 7.4, 150 mM NaCl, 10 mM $CaCl_2$, 10% $D_2O$ and 100 μM sodium trimethylsilylpropanesulfonate (DSS) as internal reference) at 310 K. The $^{1}$H-$^{15}$N HSQC pulse sequence *hsqcf3gpph19* with 128 increments and 8 scans per increment was applied for 0.1 mM $^{15}$N DC-SIGN and Langerin CRD in HBS/$Ca^{2+}$ buffer at 298 K. For screening of 650 non-fluorinated fragments of the general library against $^{15}$N LecA, a mix of 10 compounds at 1 mM each was combined with 0.15 mM $^{15}$N LecA in HBS/$Ca^{2+}$ buffer (20 mM HEPES pH 7.4, 150 mM NaCl, 10 mM $CaCl_2$, 10% $D_2O$ and 100 μM sodium trimethylsilylpropanesulfonate (DSS) as internal reference, whereas the fragments with the MBP scaffolds were validated separately at 1 mM. Similarly, 9 nonfluorinated fragments from the MBP library were screened against $^{15}$N-labeled DC-SIGN, LecA and LecB. To validate the fragment binding to proteins, we recorded $^{1}$H-$^{15}$N HSQC and TROSY NMR of proteins in the presence of DMSO as negative control and 2 mM fragments in HBS/$Ca^{2+}$ buffer. Natural ligands were used as positive controls at 1 mM MeGal, MeFuc and 30 mM D-mannose to LecA/LecB and DC-SIGN CRD/Langerin CRD, respectively. Titration experiments with $^{15}$N-labeled DC-SIGN CRD or LecB were recorded in HBS/$Ca^{2+}$ buffer with 30 mM $CaCl_2$ and 0.25 to 4–8 mM malonic acid derivatives. The $K_d$ values were calculated according to the one-site-binding model in Origin(Pro) 2020b (OriginLab Corp., USA). Ligand efficiency calculations were done according to Eq. (1) for a temperature of 298 K and 310 K for DC-SIGN CRD and LecB, respectively.

$$LE = \frac{RT \ln(K_d)}{HA},$$ (1)

with the temperature T [K], the gas constant R [kcal mol$^{-1}$ K$^{-1}$], and the number of non-hydrogen atoms HA[56].

All data were processed with NMRpipe[65] and analyzed with CcpNmr analysis[66]. For data analysis, the protein fingerprint of $^{15}$N DC-SIGN (Table S7) or Langerin CRD was assigned as reported previously[58]. The $^{1}$H-$^{15}$N TROSY resonances of $^{15}$N LecA and $^{15}$N LecB were indexed with IDs due to a lack of protein backbone resonance assignment (Tables S8-S9). Next, resonance IDs from protein spectra were transferred to the spectra obtained in the presence of compounds in order to compare the changes in chemical shift perturbations for the fast exchange peaks. The changes in chemical shift perturbations (CSPs) were calculated according to Eq. (2):

$$\triangle \delta = \sqrt{\frac{1}{2}\left[\triangle \delta_H^2 + \left(\alpha \triangle \delta_N\right)^2\right]}$$ (2)

in which δ is the difference in chemical shift (in ppm) and α is an empirical weighting factor of 0.14 for all amino acid backbone resonances[67]. The threshold value was set based on three independent measurements of reference spectra to 0.01 ppm for LecA, LecB, DC-SIGN CRD and Langerin CRD.

**Reporter-based $^{19}$F NMR**. The analogs of **1** (**5**) and **58** (**61**) were used as fluorinated spy molecules in $^{19}$F $T_2$-filtered NMR experiments. Two samples containing 0.1 mM **5** or **61** alone or a 10 μM protein in TBS/$Ca^{2+}$ buffer followed by addition of 10 mM $CaCl_2$ and the carbohydrates: 1 mM MeGal (LecA), 1 mM MeFuc (LecB/BambL) and 30 mM D-mannose (DC-SIGN/Langerin). Similarly, 0.1 mM **5** was used to prioritize hydroxamate derivatives, where 3 mM analogs of **1** were added to the samples instead of the carbohydrates. The $^{19}$F and $^{19}$F $T_2$-filtered spectra were recorded with 64 scans, a spectral width of 5 ppm, a transmitter offset at −155.5 ppm (**5**) and −155.5 ppm (**61**), the acquisition time of 0.8 s, 2 s relaxation time applying a $T_2$-filter of 384 ms. All spectra were analyzed as described above.

**PrOF NMR**. PrOF NMR was performed as reported previously[36]. Briefly, 150 μM 5FW LecA was recorded alone and in the presence of DMSO or 2–4 mM compounds in TBS/$Ca^{2+}$ buffer. The difference in chemical shift perturbations of W42 in 5FW LecA free *vs* bound forms was followed to determine $K_d$ values of compounds. The $K_d$ values were calculated according to the one-site-binding model in Origin(Pro) 2020b (OriginLab Corp., USA) from two or three independent titrations. Ligand efficiency values were calculated according to Eq. (1) for a temperature of 310 K.

**SPR**. All experiments were performed on a BIACORE X100 instrument (GE Healthcare) at 25 °C in phosphate-buffered saline (10 mM phosphate buffer pH 7.4, 2.7 mM KCl, 137 mM NaCl, 0.05% Tween-20, 100 μM $CaCl_2$, 5% DMSO). LecA was immobilized onto a CM7 chip (BIACORE) following standard amine coupling procedures. The CM7 chip was activated by three injections of a NHS/EDC mixture with a contact time of 540 s at a flow rate of 10 μL min$^{-1}$ until the response exceeded 800 RU, followed by multiple injections of LecA dissolved in 10 mM sodium acetate pH 4.5 (100 μg/mL) onto channel 2 (contact time of 540 s at

a flow rate of 10 μL min$^{-1}$). A minimum of 10,000 RU of LecA was captured onto the chip. Initial binding screens were performed with the injections of 0.2 and 1 mM of each hydroxamate (association 30 s, dissociation 60 s, 30 μL min$^{-1}$ flow rate), to identify the initial binders eliciting dose-response behaviors. Injection of the positive control (0.1 mM 4-Nitrophenyl-β-D-galactoside) was included after every forth injection cycle to monitor the activity of immobilized LecA throughout the binding screen experiments. All data evaluation was performed using BIA-CORE X100 evaluation software (v.2.0).

**Competitive fluorescence polarization (FP) assay**. The FP assay was performed in a black 384-well plate (Greiner Bio-One, Germany, 781900) with the final volume of 20 μL as described previously[53]. Briefly, 10 μL of a sample solution series (10–0.078 mM) in TBS buffer containing 10 mM Ca$^{2+}$ (TBS/Ca$^{2+}$-buffer) and 20% DMSO were added in technical triplicates to 10 μL LecA (40 μM) preincubated with a galactose-based Cy5 conjugate[68] (20 nM) in TBS/Ca$^{2+}$-buffer. Two positive controls (MeGal, $IC_{50} = 140 \pm 30$ μM and GalNAc, $IC_{50} = 1230 \pm 200$[45]), a blank (20 μM LecA, 10% DMSO in TBS/Ca2+-buffer), and a negative control (20 μM LecA, 10% DMSO, 10 nM galactose-based Cy5 conjugate in TBS/Ca2+-buffer) were included. The plate was sealed (EASYseal, Greiner Bio-One, 676,001), centrifuged (1 min, 1500 rpm, 25 °C) and incubated in a humidity chamber for 16 h. The fluorescence was measured with an excitation 590 nm and emission 675 nm filter[68] on a PheraStar FS plate reader (BMG Labtech GmbH, Germany). The blank signal of TBS/Ca$^{2+}$-buffer containing 10% DMSO was subtracted and the compounds were analyzed with the MARS Data Analysis Software (BMG Labtech GmbH Germany) using the four-parameter variable slope model. The top and bottom plateaus were defined according to the positive controls. The graphs were visualized using Graphpad Prism 5. The percentage (%) of inhibition was calculated at the highest concentration compared with 10 mM MeGal set to 100%.

**Crystallography**. Lyophilized powder of recombinant LecA was dissolved in MilliQ water containing 1 mM CaCl$_2$ to the final protein concentration of 11.7 mg mL$^{-1}$. 100 mM hydroxamate **1** stock solution in DMSO was diluted in the LecA solution to the final concentration of 20 mM and incubated at 25 °C to allow hydroxamate **1** to interact with LecA. 1.2 μL of the protein solution containing hydroxamate **1** was then mixed with 0.3 μL of LecA seed solution containing LecA microcrystals. 1.5 μL of reservoir solution (20% PEG6K, 1 M LiCl, 0.1 M sodium acetate pH 4) was added to the mixture. The entire mixture (total volume of 3 μL) was deposited on a siliconized glass slide. Crystallization was performed by the hanging drop vapor diffusion method on a 24-well plate with sealant (Hampton Research) at 19 °C. Crystals were cryo-protected in 30% PEG6000, 1 M LiCl, 100 mM sodium acetate pH 4, supplemented with 10 mM hydroxamate **1** and flash cooled in liquid nitrogen. X-ray diffraction data were collected at SOLEIL-PROXIMA2 (Saint Aubin, France) using EIGER X 9 M (Dectris) detector and analyzed as described in the Supporting Information (Table S5).

**CellFy**. The cellFy experiments using Langerin$^+$ and DC-SIGN$^+$ Raji cell lines were performed as described before[54]. Briefly, 50k cells were plated in a 96-well plate (clear, round bottom; Greiner Bio-One and mixed with varying concentrations of malonates **58**, D-mannose and 0.025 mg mL$^{-1}$ FITC-conjugated dextran (500 kDa, Sigma Aldrich) in a final volume of 50 μL following incubation for 30 min on ice. After centrifugation at 500 g for 3 min at 4 °C, the supernatant was discarded. After washing cells were treated with 50 μL 4 % paraformaldehyde (Roti-Histofix, Carl Roth) for 20 min on ice and resuspended in 100 μL fresh culture medium. The fluorescence of cells was measured by flow cytometry (MACSQuant Analyzer 16). Data was analyzed in FlowJo.

**Docking**. Docking poses were obtained by docking compound **58** in MOE[69] using the Triangle Matcher and the Rigid Receptor as placement and refinement methods, respectively. For LecA and DC-SIGN CRD, only the highest-ranking pose was taken into account, whereas for LecB the best two ones were considered. Interaction maps were also retrieved from MOE.

**Reporting summary**. Further information on research design is available in the Nature Research Reporting Summary linked to this article.

## Data availability

Relevant data are available from the corresponding author on reasonable request. Atomic coordinates for LecA bound to hydroxamic acid **35** crystal structure has been deposited in the Protein Data Bank under accession numbers 7FJH.

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

## Acknowledgements

German Research Foundation (DFG) [Ti756/5-1, RA1944/7-1] funded this work, which was in the scope of German Research Foundation and French National Research Agency [ANR-17-CE11-0048] project 'Glycomime'. We thank Max Planck Society and Olaf Niemeyer for support. Moreover, ES and CR thank a DAAD Rise Germany student J. Johnson for his contribution to screening of non-fluorinated fragments for LecA. SK and AI thank synchrotron SOLEIL (Saint Aubin, France) for access and technical support at beamline PROXIMA-2. SK thanks Annabelle Varrot for her contribution to data collection for crystallographic experiment. AI acknowledges support from Glyco@Alps (ANR-15-IDEX-02) and Labex Arcane/CBH-EUR-GS (ANR-17-EURE-0003). MHC acknowledges funding for DK-OPENSCREEN from the Ministry of Higher Education and Science (grant case no. 5072-00019B), the Technical University of Denmark, and other contributing universities.

## Author contributions

C.R. and E.S. initiated this study. E.S. performed NMR experiments. A.T. and E.S. performed the chemical synthesis and FP assay. A.I. and S.K. performed X-ray and SPR studies. D.R. and P.G. performed virtual screening. F.F. and M.D. performed CellFy and docking experiments. P.H.S. provided the infrastructure to support this study. M.H.C. supported the study with providing the 3F library. E.S. and C.R. wrote the manuscript with assistance from A.I., A.T., D.R., E.S., M.D., P.G., and SK. All authors read and approved the final version of the manuscript.

## Competing interests

The authors declare no competing interests.
