## [Peer Review File · Communications Chemistry]

Reviewers' comments:

Reviewer #1 (Remarks to the Author):

The manuscript submitted by Prof Rademacher et al. reports on the design of high affinity lectin ligands through fragments screening and targeting the binding site of lectins with non-carbohydrate moieties. While several approaches have been investigated to tackle bacterial/viral infections throughout the literature, the present study focuses on the design and evaluation of non-carbohydrate ligands of such proteins, capitalizing on a specific interaction with CA2+ necessary to some lectins. This work puts together in silico studies, along with a large set of bioanalytical techniques used to verify the affinity/interactions of the hit compounds with the targeted proteins. This is representing a huge amount of interdisciplinary work that all co-authors and PIs have been able to put together to be able to provide an in depth and comprehensive view of each type of molecules with their respective lectin.

The present study will be of extremely high utility to the glycoscience and also medicinal chemistry communities specially as it is paving the way for the design and discovery of non-carbohydrate ligands of lectins, a field that is emerging and very promising.

I recommend publication of this study with the minor comments/changes listed below.

The abbreviation MBPs for "metal-binding pharmacophores" is probably not the best one since it could also relate to "mannose-binding proteins" or "metal-binding proteins" and has been widely used for these. Although the authors clearly indicate the meaning of MBPs in their manuscript, it could be useful to try to find another abbreviation?

Table S3 about LecA ligands has many compounds (6 out of 50, ca. 10%) with "n.d." on all columns. Can the author comment on why these compounds have not been tested? If one of the reason is because another evaluation prior to this Table led to not test these compounds, then they should be simply removed from the table?

Same comment for Table S6 with 5 compounds out of 13.

Although the introduction could seem a bit long, several aspects of the field are explained and required to understand the full scope of this study, I would also appreciate a short paragraph or a few sentences about the long lasting and major (to date) approach towards high affinity lectin ligands through the design of multivalent glycoconjugates (glycopolymers, glycoclusters, glycodendrimers and so on). This can be readily achieved through the citation of a few recent reviews on the topic. Co-authors are well aware of this field having themselves published studies in this respect and they can easily select the most meaningful reviews for this.

Figures in the main text are quite heavily populated with several informations of different level in each one of them. I understand that authors have compiled in a single figure all data for a specific lectin or class of ligand studied to provide an overview in a single figure. I have been a bit confused to find quickly the information and identify easily the data (also maybe because figures appear at the end of the manuscript, I must confess). I could only suggest to divide each figure into 2 or 3 figures that would probably help the reader to have the data very close to the text and have a more efficient reading of the whole manuscript.

Reviewer #2 (Remarks to the Author):

The manuscript contains an huge amount of work, experiments, and discussion, and sometimes is difficult to follow. Generally speaking, the Figures display too much information, which is not easy to digest.

The idea of using the cation to enhance affinity is wise and the combination with the screening protocol is also excellent. One concern here is the possibility of interaction between the fragments with the isolated cation. Other possibilities could take place: as ligand-metal-ligand. The screening permits solving most of these possibilities but it is not clear to me how these possible additional interactions of the isolated cation with the fragments can be controlled.

The achieved affinities are not spectacular (mM). Although this is usual for monovalent sugar-protein interactions, the use of non-sugar ligands should overcome this low affinity. Of course, the authors are dealing with fragments and this is an initial step. Anyway, they should comment on that.

I would like to stress that the overwhelming information presented, with the use of many techniques, makes difficult the reading. The affinities are low, and this is a fact, although different techniques are employed.

In my opinion, for the screening protocol, the results/properties/affinities obtained for the diverse ligands with the lectins should be gathered in Tables, summarizing and clarifying the key points.

In contrast, the T2-filtered experiments show some results that merit explanation. For instance, in Fig 3 (panel c), the interaction of 61 with bambI seems rather strange. It looks that there is binding and that it is calcium-dependent.

Finally, the second sentence in the abstract seems to indicate that all lectin-sugar interactions are metal-dependent, which is not true.

Therefore, the manuscript presents nice science, good ideas, and interesting results. With a better presentation, the manuscript should be publishable.

Reviewer #3 (Remarks to the Author):

The paper reports for the first time that known metal binding pharmacophores (MBP) can be used as ligands for Ca-dependent lectins. This is an important result, which paves the way for a new approach to the inhibition of these difficult targets and is likely to be of high interest to a vast community of researchers. Hydroxamic acids and malonates emerge as the key hits and, despite the low affinity, selectivity among different lectins is observed. Selectivity against metalloenzymes has not been addressed, yet.

The article is well-written, but makes for difficult reading, given the very large amount of information it contains. Perhaps the section on druggability assessment could be moved to sup info, to focus directly on the MBP work, which provides the main results.

Other suggested corrections:

- the abstract, as written, seems to imply that all carbohydrate-protein interactions depend on a “central metal ion”. Similarly, at line 88 “the orthosteric site of lectins” should be “the orthosteric site of C-type lectins” Rephrase
- Figure 2f: compound numbers should be in bold

Targeting undruggable carbohydrate recognition sites through focused fragment library design

Authors: Elena Shanina, Sakonwan Kuhaudomlarp, Eike Siebs, Felix F. Fuchsberger, Maxime Denis, Priscila da Silva Figueiredo Celestino Gomes, Mads H. Clausen, Peter H. Seeberger, Didier Rognan, Alexander Titz, Anne Imberty and Christoph Rademacher

To all reviewers:

We would like to change the title of our manuscript since it emphasizes better our aim and the approach we used to address it in the manuscript.

Reviewer: 1

The manuscript submitted by Prof Rademacher et al. reports on the design of high affinity lectin ligands through fragments screening and targeting the binding site of lectins with non-carbohydrate moieties. While several approaches have been investigated to tackle bacterial/viral infections throughout the literature, the present study focuses on the design and evaluation of non-carbohydrate ligands of such proteins, capitalizing on a specific interaction with CA2+ necessary to some lectins. This work puts together in silico studies, along with a large set of bioanalytical techniques used to verify the affinity/interactions of the hit compounds with the targeted proteins. This is representing a huge amount of interdisciplinary work that all co-authors and PIs have been able to put together to be able to provide an in depth and comprehensive view of each type of molecules with their respective lectin.

The present study will be of extremely high utility to the glycoscience and also medicinal chemistry communities especially as it is paving the way for the design and discovery of non-carbohydrate ligands of lectins, a field that is emerging and very promising. I recommend publication of this study with the minor comments/changes listed below.

Comments:

The abbreviation MBPs for "metal-binding pharmacophores" is probably not the best one since it could also relate to "mannose-binding proteins" or "metal-binding proteins" and has been widely used for these. Although the authors clearly indicate the meaning of MBPs in their manuscript, it could be useful to try to find another abbreviation?

We agree with the reviewer that this abbreviation could be misunderstood by the broader readership. However, since this term was first introduced in the literature by Seth Cohen and coworkers¹, we did not impose a new name for it. For that reason, we emphasized this in the manuscript as well.

'Therefore, a target-directed FBDD campaign using metal-binding pharmacophores (MBPs) has been proposed.'

Table S3 about LecA ligands has many compounds (6 out of 50, ca. 10%) with "n.d." on all columns. Can the author comment on why these compounds have not been tested? If one of the reason is because another evaluation prior to this Table led to not test these compounds, then they should be simply removed from the table? Same comment for Table S6 with 5 compounds out of 13.

We agree with the reviewer that it requires a clarification, which we provided in the revised version of SI as follows:

Targeting undruggable carbohydrate recognition sites through focused fragment library design

Authors: Elena Shanina, Sakonwan Kuhaudomlarp, Eike Siebs, Felix F. Fuchsberger, Maxime Denis, Priscila da Silva Figueiredo Celestino Gomes, Mads H. Clausen, Peter H. Seeberger, Didier Rognan, Alexander Titz, Anne Imberty and Christoph Rademacher

Table S3

'n.d. = not determined due to poor binding of compounds in TROSY NMR or FP assay'

Table S6

'n.d. = not determined due to poor binding of compounds in TROSY or HSQC NMR'

Since these two tables serves as an overview of all tested analogs, we included two tables (1 and 2) in the revised manuscript, which should provide the summary of the key points mentioned in the manuscript.

Although the introduction could seem a bit long, several aspects of the field are explained and required to understand the full scope of this study, I would also appreciate a short paragraph or a few sentences about the long lasting and major (to date) approach towards high affinity lectin ligands through the design of multivalent glycoconjugates (glycopolymers, glycoclusters, glycodendrimers and so on). This can be readily achieved through the citation of a few recent reviews on the topic. Co-authors are well aware of this field having themselves published studies in this respect and they can easily select the most meaningful reviews for this.

We agree with the reviewer and included a few sentences on this and a reference to a review by Cecioni et al., 2015 in the revised manuscript as follows:

'Recent advances towards the development of lectin inhibitors focus on mimicking the natural ligands of lectins (glycomimetics) through the design of mono- and multivalent oligosaccharides with various conjugates (glycopeptides, glycoclusters and glycopolymers).⁴⁰ However, they pose a pharmaceutical challenge such as routes of administration and possible side effects.³⁹

Figures in the main text are quite heavily populated with several informations of different level in each one of them. I understand that authors have compiled in a single figure all data for a specific lectin or class of ligand studied to provide an overview in a single figure. I have been a bit confused to find quickly the information and identify easily the data (also maybe because figures appear at the end of the manuscript, I must confess). I could only suggest to divide each figure into 2 or 3 figures that would probably help the reader to have the data very close to the text and have a more efficient reading of the whole manuscript.

We thank the reviewer for the suggestion. We minimized the amount of information in all figures, as well as we summarized the information on tested analogs in Tables 1 and 2.

Targeting undruggable carbohydrate recognition sites through focused fragment library design

Authors: Elena Shanina, Sakonwan Kuhaudomlarp, Eike Siebs, Felix F. Fuchsberger, Maxime Denis, Priscila da Silva Figueiredo Celestino Gomes, Mads H. Clausen, Peter H. Seeberger, Didier Rognan, Alexander Titz, Anne Imberty and Christoph Rademacher

Reviewer: 2

The manuscript contains a huge amount of work, experiments, and discussion, and sometimes is difficult to follow. Generally speaking, the Figures display too much information, which is not easy to digest.

We agree with the reviewer and minimized the amount of information in all Figures, as well as moved the section on ‘druggability assessment’ to the SI in order to concise the information and focus it more on the role of MBPs in the carbohydrate-lectin interactions.

The idea of using the cation to enhance affinity is wise and the combination with the screening protocol is also excellent. One concern here is the possibility of interaction between the fragments with the isolated cation. Other possibilities could take place: as ligand-metal-ligand. The screening permits solving most of these possibilities but it is not clear to me how these possible additional interactions of the isolated cation with the fragments can be controlled.

We thank the reviewer for an excellent question, which we also have thought of. For this purpose we included a screening step that identifies Ca^{2+} /fragment interaction in the absence of protein. We argue that the chemical shift of the ^{19}F resonance was stronger perturbed in the presence of both Ca^{2+} and protein, but not in absence of CaCl_2 , which allowed us to conclude the Ca^{2+} -dependent interaction of fragments with a protein. Additionally, we clarified this in the revised manuscript as follows:

‘In the ^{19}F NMR screening of the MBP library, the chemical shifts of many ^{19}F -labeled fragments were perturbed in the presence of CaCl_2 alone, whereas even larger CSPs were observed in the presence of both CaCl_2 and the Ca^{2+} -dependent lectins as shown on the example of LecA binding to 1s-1v (Figure 1c), which allowed us to exclude direct fragment-metal interactions as a major contribution.’

The achieved affinities are not spectacular (mM). Although this is usual for monovalent sugar-protein interactions, the use of non-sugar ligands should overcome this low affinity. Of course, the authors are dealing with fragments and this is an initial step. Anyway, they should comment on that.

We agree with the reviewer, and this is certainly a focus of our future work. We commented on it in the revised manuscript as follows:

‘The analogs showed a similarly poor affinity compared to 35. Due to the small size of the fragments, there are only a limited number of the interactions they can engage in. However, they may bind tightly enough relative to their size and number of heavy atoms (HAs).¹² The ligand efficiency (LE) is the key measure relating the potency of a fragment to the number of HAs and thus, would allow the

Targeting undruggable carbohydrate recognition sites through focused fragment library design

Authors: Elena Shanina, Sakonwan Kuhaudomlarp, Eike Siebs, Felix F. Fuchsberger, Maxime Denis, Priscila da Silva Figueiredo Celestino Gomes, Mads H. Clausen, Peter H. Seeberger, Didier Rognan, Alexander Titz, Anne Imberty and Christoph Rademacher

development of drug-like molecules with enhanced binding modes for LecA (Table S3).¹³ Therefore, we selected only fragments with the highest affinity, LE and the % of inhibition values (Table 1).¹

I would like to stress that the overwhelming information presented, with the use of many techniques, makes difficult the reading. The affinities are low, and this is a fact, although different techniques are employed.

We thank the reviewer for pointing this out and we agree that the amount of techniques used appears overwhelming at the first sight. However, we want to ensure the validity of the hits identified in the experimental screening as the weak fragment-protein interactions are ideally validated in multiple orthogonal assays.¹⁴ Therefore, we included this data in the SI. We agree with the reviewer that the development of a more potent binder would have been excellent and our laboratories continue working on the improvement of the identified scaffolds.

In my opinion, for the screening protocol, the results/properties/affinities obtained for the diverse ligands with the lectins should be gathered in Tables, summarizing and clarifying the key points.

In order to minimize the amount of the information in Figures 2, 3 and 4, we summarized the data in Tables 1 and 2 in the revised manuscript.

In contrast, the T2-filtered experiments show some results that merit explanation. For instance, in Fig 3 (panel c), the interaction of 61 with bambI seems rather strange. It looks that there is binding and that it is calcium-dependent.

We agree with the reviewer that it requires a clarification. Due to the limited space in the main text, this information was clarified in the SI as follows:

'However, the 61 peak showed a stronger decrease in the presence of CaCl₂ and binding to BambL. Given the presence of the secondary sites in both lectins,¹⁵⁻¹⁷ we aimed to confirm that 61 targeted the secondary sites in both lectins. For this, we added a competitor (30 mM D-mannose and 10 mM MeFuc) expecting both ¹⁹F peaks to remain unchanged. Indeed, competitors did not influence interactions with 61, verifying its binding to a remote site in BambL and Langerin.'

Finally, the second sentence in the abstract seems to indicate that all lectin-sugar interactions are metal-dependent, which is not true.

We thank the reviewer for bringing it to our attention and rephrased this in the abstract and in the introduction as follows:

'However, its hydrophilic nature poses major limitations to the conventional development of drug-like inhibitors.'

Manuscript ID: COMMSCHEM-22-0001-T

Targeting undruggable carbohydrate recognition sites through focused fragment library design

Authors: Elena Shanina, Sakonwan Kuhadomlarp, Eike Siebs, Felix F. Fuchsberger, Maxime Denis, Priscila da Silva Figueiredo Celestino Gomes, Mads H. Clausen, Peter H. Seeberger, Didier Rognan, Alexander Titz, Anne Imberty and Christoph Rademacher

Therefore, the manuscript presents nice science, good ideas, and interesting results. With a better presentation, the manuscript should be publishable.

Targeting undruggable carbohydrate recognition sites through focused fragment library design

Authors: Elena Shanina, Sakonwan Kuhaudomlarp, Eike Siebs, Felix F. Fuchsberger, Maxime Denis, Priscila da Silva Figueiredo Celestino Gomes, Mads H. Clausen, Peter H. Seeberger, Didier Rognan, Alexander Titz, Anne Imberty and Christoph Rademacher

Reviewer: 3

The paper reports for the first time that known metal binding pharmacophores (MBP) can be used as ligands for Ca-dependent lectins. This is an important result, which paves the way for a new approach to the inhibition of these difficult targets and is likely to be of high interest to a vast community of researchers. Hydroxamic acids and malonates emerge as the key hits and, despite the low affinity, selectivity among different lectins is observed. Selectivity against metalloenzymes has not been addressed, yet.

The article is well-written, but makes for difficult reading, given the very large amount of information it contains. Perhaps the section on druggability assessment could be moved to sup info, to focus directly on the MBP work, which provides the main results.

We agree with the reviewer and moved the section on ‘druggability assessment’ to the SI as suggested.

Other suggested corrections:

- *the abstract, as written, seems to imply that all carbohydrate-protein interactions depend on a “central metal ion”. Similarly, at line 88 “the orthosteric site of lectins” should be “the orthosteric site of C-type lectins” Rephrase*

We thank the reviewer for bringing it to our attention and rephrased this in the abstract and in the introduction as follows:

‘However, its hydrophilic nature poses major limitations to the conventional development of drug-like inhibitors.’

‘However, inhibitors for the orthosteric site of Ca²⁺-dependent lectins have been a bottleneck for drug discovery.’

- *Figure 2f: compound numbers should be in bold*

We removed Figure 2f and the summary of all tested structures can be found in Table 1 (the revised manuscript) and Table S3.

Targeting undruggable carbohydrate recognition sites through focused fragment library design

Authors: Elena Shanina, Sakonwan Kuhaudomlarp, Eike Siebs, Felix F. Fuchsberger, Maxime Denis, Priscila da Silva Figueiredo Celestino Gomes, Mads H. Clausen, Peter H. Seeberger, Didier Rognan, Alexander Titz, Anne Imberty and Christoph Rademacher

References

- 1 Cohen, S. M. A Bioinorganic Approach to Fragment-Based Drug Discovery Targeting Metalloenzymes. *Accounts of Chemical Research* **50**, 2007-2016, doi:10.1021/acs.accounts.7b00242 (2017).
- 2 Chen, A. Y. *et al.* Targeting Metalloenzymes for Therapeutic Intervention. *Chemical reviews* **119**, 1323-1455, doi:10.1021/acs.chemrev.8b00201 (2019).
- 3 Ernst, B. & Magnani, J. L. From carbohydrate leads to glycomimetic drugs. *Nature reviews. Drug discovery* **8**, 661-677, doi:10.1038/nrd2852 (2009).
- 4 Cecioni, S., Imberty, A. & Vidal, S. Glycomimetics versus multivalent glycoconjugates for the design of high affinity lectin ligands. *Chemical reviews* **115**, 525-561, doi:10.1021/cr500303t (2015).
- 5 Wamhoff, E.-C. *et al.* 19F NMR-Guided Design of Glycomimetic Langerin Ligands. *ACS Chemical Biology* **11**, 2407-2413, doi:10.1021/acscchembio.6b00561 (2016).
- 6 Thépaut, M. *et al.* Structure of a Glycomimetic Ligand in the Carbohydrate Recognition Domain of C-type Lectin DC-SIGN. Structural Requirements for Selectivity and Ligand Design. *Journal of the American Chemical Society* **135**, 2518-2529, doi:10.1021/ja3053305 (2013).
- 7 Sommer, R. *et al.* Glycomimetic, Orally Bioavailable LecB Inhibitors Block Biofilm Formation of *Pseudomonas aeruginosa*. *Journal of the American Chemical Society* **140**, 2537-2545, doi:10.1021/jacs.7b11133 (2018).
- 8 Sommer, R. *et al.* Anti-biofilm Agents against *Pseudomonas aeruginosa*: A Structure-Activity Relationship Study of C-Glycosidic LecB Inhibitors. *J Med Chem* **62**, 9201-9216, doi:10.1021/acs.jmedchem.9b01120 (2019).
- 9 Meiers, J., Siebs, E., Zahorska, E. & Titz, A. Lectin antagonists in infection, immunity, and inflammation. *Current Opinion in Chemical Biology* **53**, 51-67, doi:<https://doi.org/10.1016/j.cbpa.2019.07.005> (2019).
- 10 Kuhaudomlarp, S. *et al.* Non-Carbohydrate Glycomimetics as Inhibitors of Calcium(II)-binding Lectins. *Angewandte Chemie (International ed. in English)*, doi:10.1002/anie.202013217 (2020).
- 11 Sethi, A., Sanam, S. & Alvala, M. Non-carbohydrate strategies to inhibit lectin proteins with special emphasis on galectins. *Eur J Med Chem* **222**, 113561, doi:<https://doi.org/10.1016/j.ejmech.2021.113561> (2021).
- 12 Hann, M. M., Leach, A. R. & Harper, G. Molecular complexity and its impact on the probability of finding leads for drug discovery. *Journal of chemical information and computer sciences* **41**, 856-864, doi:10.1021/ci000403i (2001).
- 13 Hopkins, A. L., Groom, C. R. & Alex, A. Ligand efficiency: a useful metric for lead selection. *Drug discovery today* **9**, 430-431, doi:10.1016/s1359-6446(04)03069-7 (2004).
- 14 Erlanson, D. A., Fesik, S. W., Hubbard, R. E., Jahnke, W. & Jhoti, H. Twenty years on: the impact of fragments on drug discovery. *Nature Reviews Drug Discovery* **15**, 605-619, doi:10.1038/nrd.2016.109 (2016).

Targeting undruggable carbohydrate recognition sites through focused fragment library design

Authors: Elena Shanina, Sakonwan Kuhaudomlarp, Eike Siebs, Felix F. Fuchsberger, Maxime Denis, Priscila da Silva Figueiredo Celestino Gomes, Mads H. Clausen, Peter H. Seeberger, Didier Rognan, Alexander Titz, Anne Imberty and Christoph Rademacher

- 15 Aretz, J. *et al.* Identification of Multiple Druggable Secondary Sites by Fragment Screening against DC-SIGN. *Angewandte Chemie International Edition* **56**, 7292-7296, doi:10.1002/anie.201701943 (2017).
- 16 Aretz, J. *et al.* Allosteric Inhibition of a Mammalian Lectin. *Journal of the American Chemical Society* **140**, 14915-14925, doi:10.1021/jacs.8b08644 (2018).
- 17 Shanina, E. *et al.* Druggable Allosteric Sites in β -Propeller Lectins. *Angewandte Chemie International Edition* **61**, e202109339, doi:<https://doi.org/10.1002/anie.202109339> (2022).

REVIEWERS' COMMENTS:

Reviewer #2 (Remarks to the Author):

The authors have satisfactorily answered all the points raised by this reviewer. Nothing else to add.